# Intestinal and Extraintestinal Pathotypes of *Escherichia coli* Are Prevalent in Food Prepared and Marketed on the Streets from the Central Zone of Mexico and Exhibit a Differential Phenotype of Resistance Against Antibiotics

**DOI:** 10.3390/antibiotics14040406

**Published:** 2025-04-16

**Authors:** Daniela Mora-Coto, Pedro Moreno-Vélez, José Luna-Muñoz, Samadhi Moreno-Campuzano, Miguel Angel Ontiveros-Torres

**Affiliations:** 1Tecnologico de Monterrey, Escuela de Ingeniería y Ciencias, Ave. Eugenio Garza Sada 2501, Monterrey, N.L., Mexico, 64849a01104714@tec.mx (P.M.-V.); 2National Dementia BioBank, AMPAEYDEN A.C., Federación Mexicana de Alzheimer, Cuautitlán Izcalli 54743, Estado de Mexico, Mexico; bnd.investigacion@gmail.com; 3Dirección de Investigación, Innovación y Posgrado, Universidad Politécnica de Pachuca, Zempoala 43830, Mexico; 4Banco Nacional de Cerebros-UNPHU, Universidad Nacional Pedro Henríquez Ureña, Santo Domingo 2796, Dominican Republic; 5Departamento de Formación Básica Disciplinaria, Escuela Nacional de Medicina y Homeopatía del Instituto Politécnico Nacional, Av. Guillermo Massieu Helguera 239, La Purísima Ticoman, Gustavo A. Madero, Ciudad de Mexico 07320, Mexico; smorenoc@ipn.mx

**Keywords:** fecal coliform bacteria, multidrug-resistant *Escherichia coli*, contaminated food

## Abstract

**Background/Objectives:** Antibiotic resistance is a serious public health problem threatening the treatment of infectious diseases caused by *Escherichia coli*, the main source of food contamination and responsible for many infectious diseases with high indices of AR profiles. Our objective was to study the presence of *Escherichia coli* in foods that are distributed and prepared on the street, characterizing its sensitivity profile and resistance to antibiotic drugs commonly prescribed in this geographical area. **Methods:** Standard procedures were performed to identify and isolate *E. coli* colonies from food samples collected during a three-year study. Susceptibility assays were conducted to determine the antibiotic resistance profile, and Colony PCR assays were performed to determine the pathogenic and antibiotic resistance genes. **Results:** A total of 189 food samples were collected, and 100% of the samples were positive for *E. coli*, with higher percentages of contamination for vegetables and fruits. ETEC (*lt*) and UPEC (*vat*, *cnf1*, *hylA*) genes were identified in 100% of the samples and DAEC (*afa*) in 27%. *E. coli* exhibited high percentages of resistance against ampicillin and amoxicillin/clavulanic acid (100%) and cephalexin (45%). The most effective antibiotics were tetracycline, TMP-SMX, polymyxin, and quinolones. The AR genes *tetA*, *sul1*, *catA1*, *strA*, *qnrS*, and *floR* were identified among the samples. **Conclusions:** Food prepared and marketed on the streets seriously threatens human health. Ampicillin and amoxicillin/clavulanic acid should not be used to treat infections caused by the multidrug-resistant ETEC and UPEC identified in this area. To our knowledge, this is the first study that explores the status of AR in this geographical area.

## 1. Introduction

Antibiotic resistance (AR) is ranked among the top ten global problems of the 21st century and has spread rapidly worldwide [1]. A comprehensive global assessment found that 5 million people died from resistant bacteria infections in 2019 [2] and projected around 10 million deaths by 2050 [3]. Of concern is the lack of surveillance and epidemiological data, which limit our understanding of the real threat posed by AR and requires a global multi-sector strategy to prevent bacterial pathogens from causing more deaths and infectious diseases in the future.

Latin America and the Caribbean reported 338,000 deaths associated with AR and 84,300 deaths attributable to this global problem [4]. Mexico has seen an increasing resistant trend against the most prescribed antibiotics that treat bacterial infections. Resistant profiles are variable among pathogens, emphasizing the importance of epidemiological surveillance throughout all Mexican states [5].

Importantly, the phenomenon of AR highly complicates the treatment of acute diarrheal diseases (ADDs) [6]. In Mexico, ADDs are among the most common syndromes of gastrointestinal infections and are the leading causes of death in children under five years of age [7]. Diarrheal diseases are ranked in the third place of the predominant causes of death [8,9] and are the second cause of disease in Mexico State.

Inadequate sanitation systems, contaminated water, and poor infection prevention strategies have been observed to encourage the spread of microbes in underdeveloped countries [10,11], common situations in food and water prepared and marketed in street conditions and other similar public places. Therefore, this type of food promotes the spread of bacteria, which may exhibit AR mechanisms [11,12].

*Escherichia coli* (*E. coli*) is a fecal coliform bacteria and a member of the Enterobacteriaceae family, responsible for more than 100 thousand deaths attributed to AR [13]. *E. coli* exhibits high indexes of multidrug resistance [4,14,15]. *E. coli* is responsible for many foodborne outbreaks and is the main source of food and water contamination [16,17].

*E. coli* is classified into several intestinal and extraintestinal pathotypes depending on its pathogenicity mechanism and site of infection. In addition, it acquires resistance genes encoding specific proteins and enzymes responsible for the effect against various antibiotic drugs. Previous studies have reported that bacteria isolated from different food sources showed the presence of enterotoxigenic *E. coli* (ETEC), diffusely adherent *E. coli* (DAEC), and uropathogenic *E. coli* (UPEC), together with antibiotic-resistant profiles against tetracycline, aminoglycosides, beta-lactams, and sulfonamides [18,19,20,21,22].

Among the few studies that explore the AR genes present in the genome of environmental *E. coli*, genes associated with quinolone resistance (*qnrA, qnrS*), sulfonamide resistance (*sul1*), chloramphenicol resistance (*catA1*), and beta-lactamase resistance (*bla_OXA_*, *bla_CTX-M_*) were detected among UPEC isolates [23,24].

Due to the serious public health problem represented by food contaminated with *E. coli* pathotypes resistant to antibiotic drugs and the lack of previous studies in the capital of the State of Mexico, the objective of our study is to evaluate the presence of *E. coli* and its possible pathotypes in fecal coliforms of food trade points in the street and to study the genotypic and phenotypic profiles of the response against antibiotics.

## 2. Results

### 2.1. Isolation of Fecal Coliform Bacteria from Food Samples

From the samples collected, 69.8% were positive for Gram-negative fecal coliform bacteria that ferment lactose, which presents a morphology of creamy fuchsia colonies in MacConkey agar (Appendix A). The colonies with those characteristics were classified into the following four categories according to the type of food from which they were isolated (Appendix A): G1, sauces made from vegetables with or without heating procedures; G2, raw vegetables; G3, whole fruits, shakes, and juices; and G4, cooked meat in different presentations and tortillas, bread, and other food made with flour that had undergone heating procedures. Each sample was reinoculated in EMB agar to confirm the fecal coliform bacteria as positive for *E. coli* due to the presence of black colonies with precipitates of metallic-green color (Appendix A). The cell wall was characterized by a Gram’s staining technique, revealing the morphology of the cells as short Gram-negative pink bacilli. Colorless colony morphologies were detected in MacConkey agar, especially in G2-22. These characteristics suggest the presence of other non-lactose-fermenting diarrheagenic pathogens, such as the *Salmonella* and *Shigella* genera (Appendix A). Further analyses are needed to demonstrate the presence of these other pathogens. In addition, *E. coli* isolation from the FCB of some samples was difficult due to other bacterial groups that exhibited pink mucoid colonies on the EMB agar (Appendix A). In these colonies, biochemical tests revealed positive saccharose, dextrose, mannitol, and citrate fermentation results. Additionally, the gas production test was positive, while the urea and indole tests were negative. These results hypothesized that the bacterial genera *Klebsiella* or *Proteus* represent a cause of food contamination in this area. Further studies are needed to corroborate these findings. Only the metallic green-black colonies (*E. coli*) were included for further analysis. However, an enormous degree of contamination with various pathogens was observed in Groups 1, 2, and 3 in all years, while Group 4 had the lowest percentage of *E. coli* contamination. A total of 54%, 67%, and 70% of the samples collected in 2021, 2022, and 2023, respectively, were positive for *E. coli* (Figure 1a). These results exhibited high percentages of *E. coli* contamination in food prepared and marketed on the streets with variations from one type of food to another; G4 is the one with the lowest percentages of contamination, while G3, G2, and G1 exhibited the highest percentages. These results indicate that heating procedures increase food quality (Figure 1b).

### 2.2. Pathogenic Genes Identification

Using the polymerase chain reaction of colony and horizontal electrophoresis in 2% agarose gels, the presence of amplicons for genes corresponding to intestinal and extraintestinal pathotypes of *E. coli* was explored. Our study shows variation in the presence of pathogenic genes between years and food types. Notably, UPEC, with at least one pathogenic gene detected in each sample, emerged as one of the most frequently identified pathotypes in food.

Three pathogenicity genes of UPEC were explored as follows—*vat*, *cnf1*, and *hlyA*—with amplicons whose molecular weights are 289, 498, and 1177 base pairs, respectively. The amplicon for the *vat* gene was detected in all of the samples except the G3-23. In the case of the *cnf1* gene, ten samples were positive for the amplicon, with the G4-23 not presenting the amplicon of the coding gene for the cytotoxic necrotic factor 1. Of the virulence factors of UPEC, the *hlyA* gene had a poor presence in the food samples. Only G3-22 shows the amplicon of alpha-hemolysin. In addition to UPEC, ETEC was detected among all the samples by the identification of the heat-labile toxin. All samples show the amplicon of the *lt* gen. Furthermore, from the year 2021, DAEC was detected in G1-21, G2-21, and G4-21 using the amplicon for the *afa* gene (Appendix A). All samples exhibited the presence of the *E. coli* pathotypes ETEC and UPEC, with the amplification of at least two pathogenic genes. Only Group 3 and Group 4 exhibited the presence of three pathotypes: ETEC, UPEC, and DAEC. In addition, G3-22 was the sample that tested positive for the three virulence factors of UPEC. G2-21 and G4-23 exhibited only one of the virulence factors of UPEC (Table 1).

### 2.3. Antibiotic Susceptibility

Two previously reported protocols for bacterial inoculum preparation were tested to explore the antibiotic-resistant phenotype. Notably, the areas of inhibition generated by antibiotic drugs versus bacteria were different between the two protocols. For example, for areas of inhibition with tetracycline and cephalexin, the results of the TSB medium were significantly higher than those of the saline solution. On the contrary, inhibition areas from SS with ampicillin and chloramphenicol were significantly higher than those of TSB (*p* ≤ 0.05 value). For the ampicillin, amikacin, and nalidixic acid results, no area of inhibition in bacterial growth was observed in the TSB medium (Figure 2). These results showed that *E. coli* isolated from food in this geographical area may have a differential behavior to the microbicidal effects of antibiotics. The response depends on the medium in which it is grown for sensitivity phenotype trials and the antibiotic tested. Taking this reference effect and noting that the broth will offer higher nutritional conditions, the TSB medium was selected to perform subsequent experiments.

The sensibility analysis showed that *E. coli* isolated from the samples are highly resistant to cephalexin (45%), ampicillin (100%), and amoxicillin (100%). Importantly, 45% of the samples also showed intermediate resistance to cephalexin. On the other hand, tetracycline, amikacin, polymyxin, levofloxacin, and ciprofloxacin exhibited 100% effectiveness since all the samples were susceptible to the above-mentioned antibiotics. Chloramphenicol and nalidixic acid showed a higher percentage of susceptibility. However, some of the samples displayed intermediate resistance. Finally, it should be noted that although the percentage of contamination in 2023 increased compared to 2021 and 2022, the incidence of resistance was similar between the years (Table 2).

The sensibility response against the twelve antibiotics, which are common clinical prescription antibiotics, in the central zone of Mexico is shown (Figure 3). The values obtained from the sensibility assays were adjusted according to the higher value of the standard established by the CLSI, and the responses were compared between samples of the same year. Importantly, the isolated *E. coli* responded similarly against amoxicillin, gentamicin, and TMP-SMX, independently of the food type and the year of sampling.

G1 exhibited significant differences from the other groups within the zones of inhibitions obtained against chloramphenicol, nalidixic acid, levofloxacin, and ciprofloxacin (Figure 3f–i). At least in one year of sampling, G1 was significantly different from the results obtained in G2 in seven of the analyses (Figure 3a–c,e–h) and from the G3 results in seven of the analyses (Figure 3a,b,d,e,g–i). G2 was significantly different from G3 in the analyses against ampicillin, amikacin, and polymyxin (Figure 3c–e) and from G4 in the analyses against levofloxacin and ciprofloxacin (Figure 3f,g). Finally, G3 was significantly different from G4 in the analysis of tetracycline, polymyxin, nalidixic acid, and chloramphenicol (Figure 3a,e,h,i).

The inhibition zones between the antibiotics studied were compared by the group samples within the same year (Figure 4). The analysis of G1 shows significant differences between AMP and AMX with the rest of the antibiotics (Figure 4a). This pattern continues in the analysis of G2 (Figure 4b), G3 (Figure 4c), and G4 (Figure 4d). All four comparisons exhibited similar results. AMP, AMX, and, in a lower incidence, CEP, exhibited no significant differences but were significantly different from the results of the other antibiotics. Additionally, these three antibiotics show lower inhibition values when compared to POL and TMP-SMX, which displayed higher inhibition values, and, in almost all, the analyses were significantly different from the other antibiotics. In most cases, TET, AMK, LEV, CIP, NA, CLO, and GEN show no significant differences.

### 2.4. Antibiotic-Resistant Genotype

As in Section 2.2, the antibiotic-resistant genes were explored by Colony PCR, and amplicons were visualized by electrophoresis in 2% agarose gels. The results exhibited the presence of multidrug-resistant *E. coli* among the samples studied. The amplicons of the genes *strA*, *sul1*, *catA1*, *floR*, *qnrS*, and *tetA* with a size of 547, 698, 547, 398, 428, and 210 bp, respectively, demonstrated the presence of antibiotic-resistant mechanisms against streptomycin (*strA*), sulfonamides (*sul1*), chloramphenicol (*catA1*, *floR*), quinolones (*qnrS*), and tetracycline *(tetA*) (Appendix A). The *floR* gene encodes for the efflux pump responsible for exporting chloramphenicol outside the cytoplasm. It was detected in all samples from 2022 and 2023, except for G3-22 and G3-23. The gene *strA*, which encodes the aminoglycoside-phosphotransferase enzyme, was detected only in G3-21, G3-22, and G4-22. However, none of the 2023 samples amplified this gene. In 2021 and 2022, the *sul1* gene and *catA1* gene were detected in all samples. However, only G1-23 exhibited the amplicon of the gene *catA1*. The former encodes for chloramphenicol acetyltransferase, while *sul1* encodes for the sulfonamide-resistant dihydropteroate synthase. The gene *tetA* was detected only in samples from 2021, encoding for an efflux pump. Finally, the quinolone-resistant mechanism (*qnrS*) was only detected in Group 1 and Group 4 (Table 1).

## 3. Discussion

In recent years, it has been documented that diarrheal diseases in the State of Mexico are transmitted mainly by contaminated food and water [25]. There is a large consumption of food prepared and marketed on the streets due to its low cost. *E. coli* and other fecal coliform bacteria are mostly isolated from raw fruits and vegetables, indicating fecal contamination. Two studies of freshly squeezed orange juice exhibited the presence of *E. coli* in 8.3% [26] and 14% [27] of the samples. Additionally, mung bean sprouts, serrano peppers, and jalapeño peppers also showed contamination by *E. coli* in 95% [28], 58%, and 38% [29], respectively. Additionally, ETEC was present in 12% and 2% of serrano pepper and jalapeño pepper samples, respectively. Another study demonstrated poor hygienic manipulation of coriander in Mexico with the presence of *E. coli* in 43% of the samples, conjointly with multidrug-resistance profiles [30].

ETEC and UPEC pathotypes were also isolated from fresh cheese samples from retail markets in Mexico [22,31]. Prickly and coriander samples also exhibited the presence of the STEC, EPEC, and ETEC pathotypes [20,30]. Another study in Tabaco reported the presence of STEC, DAEC, and UPEC in fresh cheese [22]. In Pachuca, a study reported the presence of ETEC, EIEC, and STEC in mung bean samples [28].

Previous findings correlated with the results of this study, showing that raw vegetables and fruits represent a threat to human health. In our research, G3, G2, and G1 exhibited the highest percentage of *E. coli* contamination, groups that include sauces, vegetables, and fruit and juices. All of these types of food lack heating procedures, like steaming, baking, and frying, among others. These findings demonstrate poor, inadequate manipulation during food preparation due to the bacterial content.

UPEC and ETEC were the most prevalent extraintestinal and intestinal pathotypes found in our study, which may be due to the proximity to the Lerma River of the study areas. However, a few studies explore the relationship between diseases transmitted by this type of food distributed by street vendors and consumption in the population or its impact on public health.

Although the *E. coli* pathotypes identified in previous studies are different (and can be explained by the years after the coronavirus pandemic and the earthquake in September 2017 in Mexico), there is an evident problem of contamination among vegetables and fruit products that represents a threat to public health.

UPEC is responsible for 70% to 95% of all cases of urinary tract infections (UTIs). Importantly, 40–50% of women suffer at least one UTI during their lifetime [32]. The toxins alpha-hemolysin (HlyA), vacuolating autotransporter toxin (Vat), and cytotoxic necrotic factor 1 (CNF1) are highly studied toxins associated with UPEC pathogenicity. The first one is a calcium-dependent toxin that mediates host apoptosis and inflammatory pathways. Vat is related to cytopathic effects, such as vacuolation and swelling, and CNF1 mediates ubiquitination and proteasomal degradation [33]. UTIs are mainly treated with antibiotics, and the presence of multidrug-resistant UPEC among the food samples represents a serious health threat to UTI treatments. Based on the clinical guidelines, the most common antibiotics used to treat UTIs are beta-lactams, quinolones, cephalosporins, penicillin, and aminoglycosides [34].

ETEC is one of the main causative agents of diarrhea in children and is frequently isolated from acute diarrhea cases. The main mechanism of pathogenicity is the secretion of thermolabile and thermostable toxins, which increase the concentrations of cyclic nucleotides, triggering the opening of channels of chloride anions by phosphorylation, causing osmotic diarrhea by the exit of water and ions toward the intestinal lumen [35]. Studies have focused on analyzing the presence of ETEC among isolates from infantile patients, and resistance has been documented mainly against ampicillin, cephalosporines, nalidixic acid, tetracycline, and trimethoprim-sulfamethoxazole [36,37,38,39]. However, quinolones and fluoroquinolones are the most common antibiotics used to treat diarrheal diseases [40].

It is noted that antibiotic resistance has become a national problem. The resistance profiles reported around Mexican regions differ. Gomez-Aldapa and colleagues found high resistance against amikacin and gentamicin among the *E. coli* samples isolated from prickly pears and coriander [20,30]. *E. coli* isolated from water samples exhibited resistant profiles against tetracycline, chloramphenicol, and TMP-SMX [41]. In Tamaulipas, *E. coli* was also reported to be resistant to cephalosporins, beta-lactams, and tetracycline [42]. Fresh cheese samples in Pachuca were analyzed, and the isolated *E. coli* exhibited resistance against amoxicillin/clavulanic acid and amikacin [31]. In this study, *E. coli* was highly resistant to ampicillin, amoxicillin/clavulanic acid, and cephalexin, while TMP-SMX, tetracycline, and amikacin inhibited the bacterial growth effectively. It has been noted that bacteria isolated from other areas of Mexico displayed resistance profiles against the most effective antibiotics of this study. These findings highlight the importance of constant surveillance of the antibiotic resistance status of each Mexican area.

Studies from other Latin American regions exhibited differential antibiotic-resistant profiles. For example, in Portugal, *E. coli* strains isolated from food items displayed resistant profiles against ciprofloxacin, nalidixic acid, chloramphenicol, tetracycline, and TMP-SMX [43], antibiotics to which the *E. coli* strains from this study were susceptible. Likewise, Colombian Paipa cheese exhibited the presence of *E. coli* in 25% of the samples with sensitivity profiles against ampicillin, while it revealed resistant profiles to TMP-SMX [44]. These results demonstrate the importance of national and local strategies to counter this problem.

G1 responded differently to the antibiotics than *E. coli* isolated from vegetables, fruits, and cooked items. It is noticed that in five of the graphs, no significant differences were found in 2023. Here, the isolated *E. coli* responded similarly to the antibiotics even when the contamination percentage was higher and the incidence of resistance in 2023 was similar to the other years. G1 includes sauces made from vegetables with or without heating procedures.

Finally, ampicillin and amoxicillin showed lower values of inhibitions and were significantly different from the other commonly prescribed antibiotics. This result correlates with the above findings and demonstrates the inefficiency of these drugs as treatments for the *E. coli* present in this geographic area. On the contrary, TMP-SMX and POL displayed higher inhibition values, also significantly different from the other antibiotics, proven to be effective antibiotics to treat infections caused by *E. coli*.

A few studies in Mexico have attempted to identify antibiotic-resistant genes present in the *E. coli* genome isolated from food and or to produce a correlation against the resistance phenotype response. Antibiotic resistance genes reported in Mexican studies include *tetA* and *tetB, strA* and *strB*, *sul2*, and *sul3*, together with beta-lactamase genes (*bla_TEM_*, *bla_CTX-M_*, *bla_OXA_*, and *bla_SHV_*) and quinolone resistance genes (*qnrA*, *qnrB*, and *qnrS*) [23,24,42,45]. Additionally, polymyxin was considered the last treatment option due to the increasing resistance to the first line of antibiotics. However, a previous study found the *mrc-1* gene on a fecal sample, a gene associated with polymyxin resistance [46]. Thus, the absence of resistance against polymyxin in this study is important.

The molecular mechanisms of antibiotic drug resistance that we can detect by the amplicons obtained from the DNA of *E. coli* involve transmembrane proteins of the cell wall, structural modifications or incorporation of functional groups by enzymatic action, and the synthesis of new proteins. The *tetA* gene encodes for a Major Facilitators Superfamily protein, an efflux pump that binds to tetracycline and exports it to the outside using an antiport system. The *sul1* gene encodes the new sulfonamide-resistant enzyme dihydropteroate synthase, responsible for sulfonamide resistance. The *strA* gene encodes for the enzyme aminoglycoside-phosphotransferase, responsible for phosphorylating aminoglycoside antibiotics in the 3′-hydroxyl group, decreasing its affinity for the 30S subunit of the ribosome. *QnrS* encodes a protein that confers reduced susceptibility to quinolones. Finally, the *catA1* gene encodes the enzyme chloramphenicol-acetyltransferase responsible for acetylating these antibiotics and blocking their interaction with ribosomal 50S subunits [47,48] (Appendix A).

We found similarities between the previously mentioned studies and the amplified genes in this study. However, the AR genotype does not always correspond to the AR phenotype. The *tetA* (to mention one) indicates resistant mechanisms to tetracycline. Although the sensibility assay showed results of susceptibility to this antibiotic, environmental factors play a crucial role in gene expression, and these findings denote the importance of the environment as a route and source of antibiotic resistance spread. However, how environmental factors impact gene expression is poorly studied and understood [49].

While this work provides important information regarding the resistance behavior of bacteria present in food prepared and marketed under street conditions, there are some limitations that must be considered. Due to the dynamic factor of this type of food, it was difficult to collect samples from the same seller during the three years that our study lasted. There is a socio-economic impact that affects the presence or absence of vendors. However, we offer an approximation of the resistance phenomenon in this geographical area, collecting samples during the same season and at the same points of sale. Therefore, we provide a first insight into the state of resistance and its correlation with microbial load in food, laying the groundwork for further research.

## 4. Materials and Methods

### 4.1. Food Sampling and Bacterial Isolation

One hundred and eighty-nine samples of food prepared and marketed in street conditions were collected from 2021 to 2023 in three defined geographical locations and during the same season. The three selected zones were located at “Mercado 16 de septiembre” (19.295329720085046, −99.65233073060944), “Mercado Juárez” (19.278670344550665, −99.6390527898009), and “Terminal de Autobuses” (19.277737071702695, −99.64292748169059), which are crowded areas with several food stalls by the roadside. The food items were selected aleatory from the stalls that matched the following characteristics: ready-to-eat food, exposure to vehicular traffic, and lack of proper infrastructure, like access to safe water and refrigeration. Each food was scraped using a bacteriological scraper, and the samples were plated on MacConkey agar (BIOXON), which is selective for lactose-positive Gram-negative fecal coliform bacteria. The colonies obtained with round morphology and fuchsia color, characteristic of *E. coli* growth, were reinoculated on Eosin Methylene Blue agar (EMB, BIOXON), another selective medium for *E. coli* colonies. Dark colonies with metallic precipitation were selected as *E. coli*-positive samples. The samples collected were grouped into four food types as follows: Group 1 (G1), sauces made from vegetables with or without heating procedures; Group 2 (G2), raw vegetables; Group 3 (G3), whole fruits, shakes, and juices; Group 4 (G4), cooked meat in different presentations and tortillas, bread, and other food made with flour that had undergone heating procedures. The Raosoft sample size calculation method with a 5% margin of error and 95% confidence level was used to determine the sample size.

### 4.2. Biochemical Tests and Characterization of the Bacterial Wall by Gram Staining

Isolated bacteria were subjected to biochemical tests following the standard methods. The biochemical tests included sugar fermentation and indole tests, jointly with the implementation of triple sugar iron (TSI) agar and urea agar [50,51]. Additionally, a sample was taken from the positive samples using a bacteriological loop, and the Gram stain protocol described by the American Society for Microbiology was followed [52].

### 4.3. Antibiotic Susceptibility Testing

The susceptibility of *E. coli* to antibiotic drugs was explored following the Kirby−Bauer method established by the Clinical Laboratory Standard Institute (CLSI) [53]. The CLSI reported two experimental protocols for inoculum preparation: the growth method using trypticase soy broth (TSB) and the direct colony suspension method using a 0.45% saline solution (SS) to adjust the bacterial suspension density [54]. Both protocols were tested, looking for the one that offered a better response against antibiotics. To analyze the resistance phenotype of the isolated bacteria, Muller Hinton agar (BIOXON) plates were reinoculated using a sterile swab, and absorbent discs (Whatman paper No. 3) with a standard concentration of antibiotics were placed on the inoculated plate and subsequently incubated at 37 °C for 24 h. The discs were prepared following the methodology of Vineetha et al. [55]. Twelve antibiotics were used at concentrations established by the Clinical Laboratory Standards Institute (CLSI): tetracycline (TET, 30 µg), cephalexin (CEP, 30 µg), sulfamethoxazole/trimethoprim (TMP-SMX, 1.25 µg/25 µg), ampicillin (AMP, 10 µg), amikacin (AMK, 30 µg), polymyxin B (POL, 30 µg), levofloxacin (LEV, 5 µg), ciprofloxacin (CIP, 5 µg), nalidixic acid (NA, 30 µg), chloramphenicol (CLO, 10 µg), amoxicillin/clavulanic acid (AMX, 20 µg/10 µg), and gentamicin (GEN, 10 µg). The diameters of the inhibition areas were measured with a vernier and interpreted as resistant (R), intermediate (I), and susceptible (S), following the CLSI [56].

### 4.4. Cell Lysis

Plasmid and chromosomal DNA were obtained from *E. coli*-positive colonies following heat shock and osmotic stress protocols for cell lysis. Three colonies were picked with a bacteriological loop and resuspended in 1 µL of MilliQ deionized water (Milli-Q System, Millipore, Darmstadt, Germany). The suspension was mechanically shaken using a vortex and subjected to boiling in a water bath for 1 min, after which it was immediately cooled by placing the tubes on ice. Finally, the samples were stored at −70 °C for further use as template DNA in Colony Polymerase Chain Reaction (PCR) experiments.

### 4.5. Colony PCR

*E. coli* pathogenicity and antibiotic-resistant gene identification were implemented using a previously reported Simplex PCR protocol [57]. The primers used for the amplification of ten pathogenic genes of *E. coli* and the resistance genes against tetracycline (*tetA*), sulfonamides (*sul1*), chloramphenicol (*cat1*, *floR*), beta-lactams (*bla-tem*, *oxa48*), streptomycin (*strA*), colistin (*CLR5*), fluoroquinolones, and quinolones (*qnrS*) in the *E. coli* genome are described in Appendix A, respectively. The components of the reaction mixture adjusted to a total volume of 25 µL include 2.5 µL of PCR buffer minus MgCl_2_ (20 mM Tris-HCL (pH 8.4), 50 mM KCl), 1.5 mM MgCl_2_, 2.0 µL of 10 mM dNTPs mixture (0.2 mM each), 0.1 µL of Taq DNA polymerase (5U/µL) (Thermo Fisher Scientific, Waltham, MA, USA), 2.0 µL of primer mix (0.8 µM/µL), 2.0 µL of DNA suspension, and 15.4 µL of Milli Q water (sterile and deionized). The condition for each protocol is detailed in Appendix A. Aliquots of 2 µL of the amplicons with 8 µL of the loading buffer (DNA electrophoresis Sample Loading Dye) were analyzed on a 2% horizontal agarose gel for 70 min at 80 V, 400 mA in 1X TAE (Tris-Acetate EDTA) buffer and stained with SYBR Safe 3X. The gels were analyzed using a 100 bp DNA Ladder Marker (Thermo Fisher Scientific) and a gel documentation system (Bio-Rad Gel Doc XR+ System, Hercules, CA, USA).

### 4.6. Statistical Analysis

A t-student test was conducted to compare the two protocols for inoculum preparation. The confidence intervals were 95%, and the significance level was set at *p* < 0.05. The data were analyzed using the Minitab software (ver. 21.4.0.0). The one-way ANOVA through the Tukey test was used to compare the susceptibility response between the groups against each antibiotic. The significance level was set at *p* < 0.001, and the data were analyzed using the JASP statistical software (ver. 0.18.3.0). The results from the same years between antibiotics were compared by group through a Fisher test LSD with a confidence interval of 95% using the software Statgraphics (19.6.05).

## 5. Conclusions

In this study, we found high percentages of *E. coli* contamination in food, with a trend of constant increase over the years. This represents a serious health risk for people who consume food prepared and marketed on the streets in developing countries such as Mexico. It should also be noted that pathogenic genes associated with the UPEC, ETEC, and DAEC pathotypes were detected. These important pathogens are responsible for recurrent urinary tract infections, such as acute diarrhea, infections commonly treated with antibiotics. In addition, *E. coli* strains isolated from food samples exhibit multidrug-resistance profiles, especially against ampicillin and amoxicillin/clavulanic acid, while the drugs that showed greater effectiveness were polymyxin and TMP-SMX. This result may be important for local health centers to identify the need to more accurately restructure therapeutic intervention against infections caused by resistant *E. coli* in this geographical area. Furthermore, the phenotype and the genotype are not always correlated, indicating that several factors may regulate the AR mechanisms. These findings provide important information to local health centers, allowing medical specialists to analyze and determine better therapeutic strategies and highlighting the importance of strict hygienic regulations on food prepared and marketed on streets and drug prescription control.

## Figures and Tables

**Figure 1 antibiotics-14-00406-f001:**
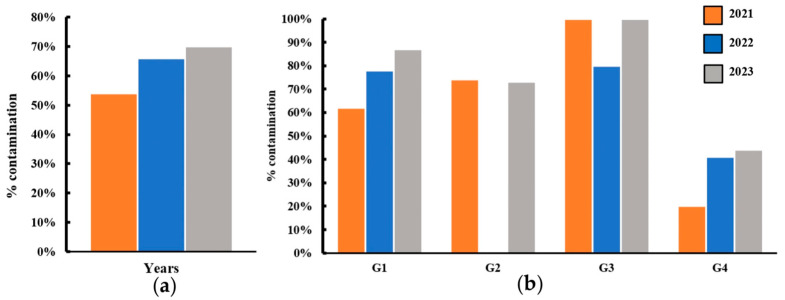
Percentage of *E. coli* contamination per year and by group per year. (**a**) The percentage of *E. coli* contamination from 2021 to 2023. The total number of samples collected each year was considered 100%. (**b**) The percentages of *E. coli* contamination by group. Each year, the samples collected were classified into four groups based on Appendix A, and the total samples of each group were considered as the 100%. G1: group of sauces; G2: group of vegetables; G3: group of fruits/juices/shakes; G4: group of meat- and corn-based foods.

**Figure 2 antibiotics-14-00406-f002:**
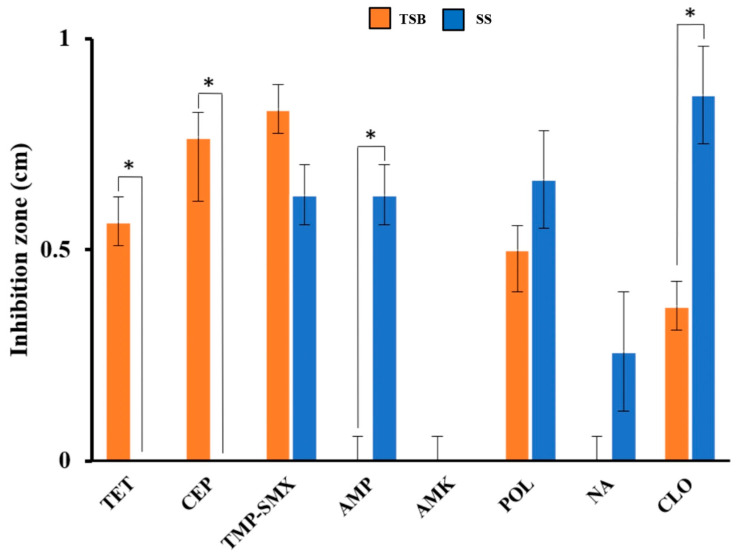
Differences in the diameter of the inhibition zone between different protocols for bacterial inoculum preparation. The effectiveness of sensitivity against *E. coli* antibiotic drugs in tryptic soy broth (TSB) and saline solution (SS) was compared using the antibiotics TET, CEP, TMP-SMX, AMP, AMK, POL, NA, and CLO. Experiments were carried out in triplicates, and standard deviations are displayed as the error bars shown in the bars. Statistical differences were calculated and are highlighted as * *p* ≤ 0.05.

**Figure 3 antibiotics-14-00406-f003:**
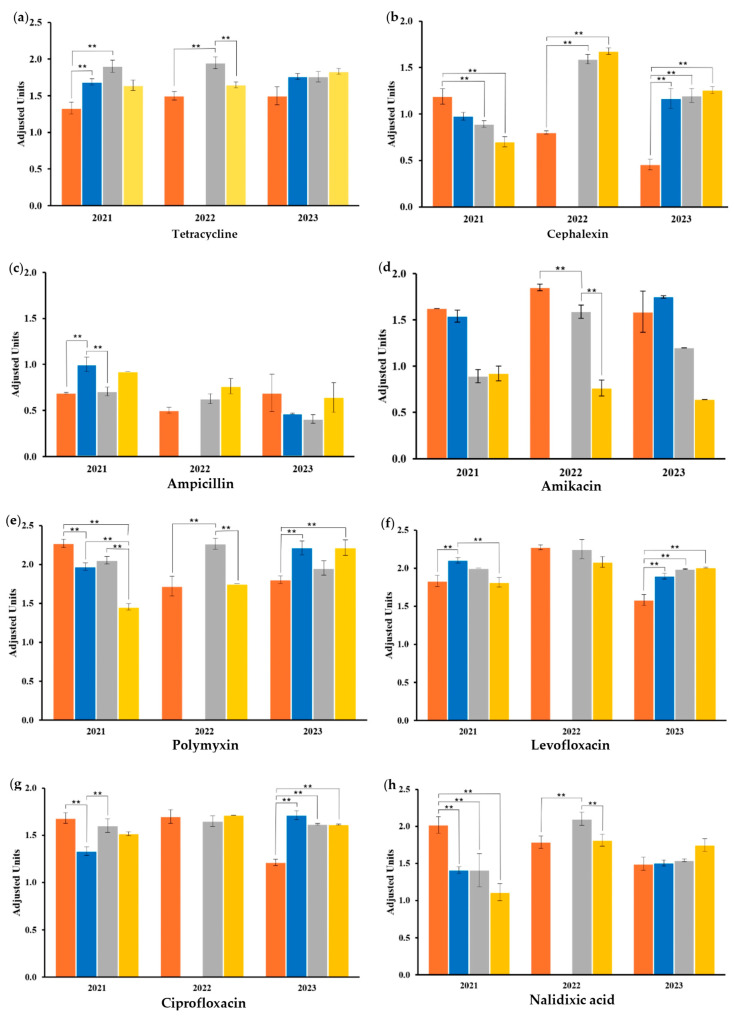
Evaluation of the sensitivity response of *E. coli* isolated from the different food groups against the antibiotics in the three years of the study. Results are shown by antibiotic (**a**) tetracycline, (**b**) cephalexin, (**c**) ampicillin, (**d**) amikacin, (**e**) polymyxin, (**f**) levofloxacin, (**g**) ciprofloxacin, (**h**) nalidixic acid, and (**i**) chloramphenicol. Experiments were carried out in triplicates, and the standard deviation is displayed as the error bars shown in the graphs. Statistical differences were calculated and are highlighted as ** *p* ≤ 0.001.

**Figure 4 antibiotics-14-00406-f004:**
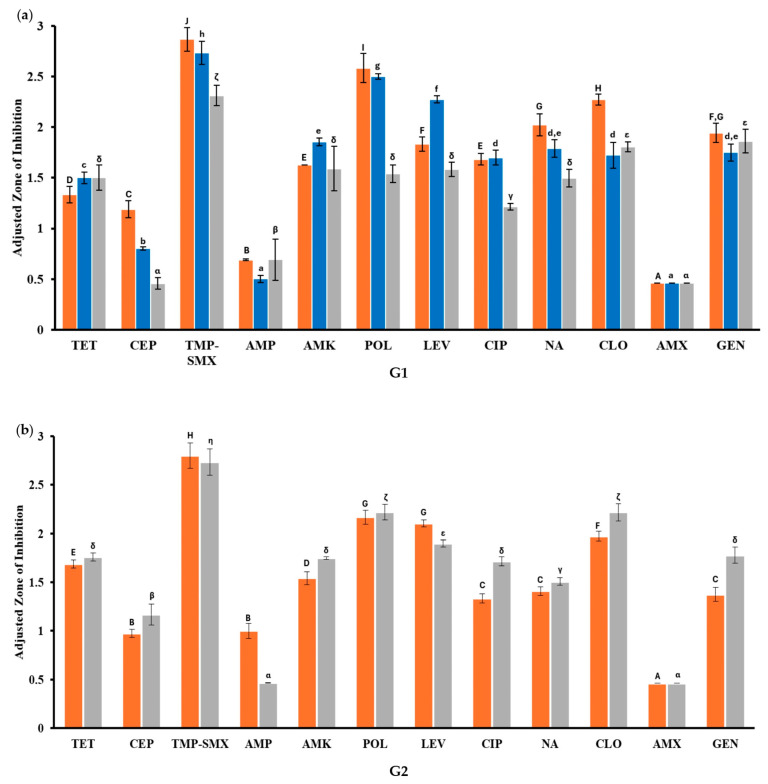
Evaluation of the inhibition results between the antibiotics of the same year by group sample. Each graph shows the analyses of one of the groups, and the analysis was carried out by year. Results are shown by group sample, (**a**) group 1, (**b**) group 2, (**c**) group 3, and (**d**) group 4. Experiments were performed in triplicates, and the standard deviation is displayed as the error bars shown in the graphs. Statistical differences were calculated and are highlighted as 2021: (A, B, C, D, E, F, G, H, I, J); 2022: (a, b, c, d, e, f, g, h); 2023: (α, β, γ, δ, ε, ζ, η, θ) *p* ≤ 0.05.

**Table 1 antibiotics-14-00406-t001:** Integration of the identified genes (pathogenicity and resistance) and the multidrug-resistance phenotype in the food groups studied per year in the isolates of *E. coli*.

Sample	*E. coli* Pathotypes	Pathogenic Genes	AR Genes	AR Phenotype
G1-21	ETEC, UPEC, DAEC	*lt*, *afa*, *cnf1*	*tetA*, *sul1*, *catA1*	R: AMP, NA, AMX I: CEP
G2-21	UPEC, ETEC	*lt*, *afa*, *vat*	*tetA*, *sul1*, *catA1*	R: CEP, AMP, AMX I: NA, GEN
G3-21	ETEC, UPEC, DAEC	*lt*, *cnf1*, *vat*	*tetA*, *sul1*, *catA1*, *strA*	R: CEP, AMP, CIP I: NA
G4-21	ETEC, UPEC, DAEC	*lt*, *afa*, *cnf1*, *vat*	*tetA*, *sul1*, *catA1*	R: CEP, AMP, AMX I: NA, CLO, GEN
G1-22	ETEC, UPEC	*lt*, *cnf1*, *vat*	*tetA*, *sul1*, *catA1*, *qnrS*	R: CEP, AMP, AMX
G3-22	ETEC, UPEC	*lt*, *hlyA*, *cnf1*, *vat*	*tetA*, *sul1*, *catA1*, *strA*, *floR*	R: AMP, AMX I: CEP, AMK
G4-22	ETEC, UPEC	*lt*, *cnf1*, *vat*	*tetA*, *sul1*, *catA1*, *strA*, *floR*, *qnrS*	R: AMP, AMX
G1-23	ETEC, UPEC	*lt*, *cnf1*, *vat*	*catA1*, *floR*	R: CEP, AMP, AMX
G2-23	ETEC, UPEC	*lt*, *cnf1*, *vat*	*floR*	R: AMP, AMX I: CEP
G3-23	ETEC, UPEC	*lt*, *cnf1*, *vat*		R: AMP, AMX I: CEP
G4-23	ETEC, UPEC	*lt*, *cnf1*	*floR*	R: AMP, AMX I: CEP

ETEC: enterotoxigenic *E. coli*; UPEC: uropathogenic *E. coli*; DAEC: diffusely adherent *E. coli*. *lt*: thermolabile toxin; *afa*: afimbrial adhesin; *cnf1*: cytotoxic necrotic factor 1; *vat*: vacuolating autotransporter toxic; *hlyA*: alpha-hemolysin toxin.

**Table 2 antibiotics-14-00406-t002:** Sensitivity analysis of the isolated *E. coli* against the twelve antibiotics included in this study. The inhibition zones were interpreted based on the Clinical Laboratory Standard Institute protocol and classified as follows: **R:** resistant; I: intermediate; S: susceptible.

Antibiotic	G1-21	G1-22	G1-23	G2-21	G2-23	G3-21	G3-22	G3-23	G4-21	G4-22	G4-23
TET	S	S	S	S	S	S	S	S	S	S	S
CEP	I	**R**	**R**	**R**	I	**R**	I	I	**R**	S	I
TMP-SMX	S	S	S	S	S	S	S	S	S	S	S
AMP	**R**	**R**	**R**	**R**	**R**	**R**	**R**	**R**	**R**	**R**	**R**
AMK	S	S	S	S	S	S	I	S	S	S	S
POL	S	S	S	S	S	S	S	S	S	S	S
LEV	S	S	S	S	S	S	S	S	S	S	S
CIP	S	S	S	S	S	S	S	S	S	S	S
NA	**R**	S	S	I	S	I	S	S	I	S	S
CLO	S	S	S	S	S	S	S	S	I	S	S
AMX	**R**	**R**	**R**	**R**	**R**	**R**	**R**	**R**	**R**	**R**	**R**
GEN	S	S	S	I	S	S	S	S	I	S	S

## Data Availability

The original contributions presented in this study are included in the article/Appendix A. Further inquiries can be directed to the corresponding author.

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
