# Peer review of "Intestinal and Extraintestinal Pathotypes of Escherichia coli Are Prevalent in Food Prepared and Marketed on the Streets from the Central Zone of Mexico and Exhibit a Differential Phenotype of Resistance Against Antibiotics"

_antibiotics, 2025, doi:10.3390/antibiotics14040406_

Round 1

Reviewer 1 Report

Comments and Suggestions for Authors

In this body of work, the authors have characterized the pathotypes of E. coli prevalent in street side food from central zone of Mexico. The isolates were grouped and studied for AR using disc assays as well as colony PCR to identify genetic determinants of resistance.

Line 51: no need of "from"

Line 59: Does this pertain to global data? please clarify and include the reference

It is crucial to introduce the specificities of Groups 1,2,3, and 4 in the result section and not just in the figure legend of fig. 1. Co-relation between heating and the findings of Fig. 1 need to be explained better.

In Fig. 2: It would be better to address the inhibition halos as zone of clearance or zone of inhibition. Given that the authors are using the unit of "cm" to measure inhibition, it should be addressed as diameter and not area; especially when cm is used as the unit

Throughout the draft, "sensibility" can be replaced with "sensitivity" to improve the scientific readability of this manuscript.

Overall, the manuscript is very dense on bar graphs and other graphically appealing ways can be used to present this data.

Please also outline the limitations of this study. 

Author Response

Thank you very much for taking the time to review this manuscript. Please find the detailed responses below and the corresponding revisions in the re-submitted files.

Comments 1: Line 51: no need of "from"

Response 1: Thank you for pointing this out. In response, we have removed “from” from line 51.

Comments 2: Line 59: Does this pertain to global data? please clarify and include the reference

Response 2: Thank you for the observation. Accordingly, we have modified that sentence as follows (Line 58-60): “In Mexico, ADDs are among the most common syndromes of gastrointestinal infections and are the leading causes of death in children under five years of age [7]”. We added one reference and modified the reference numbers.

Comments 3: It is crucial to introduce the specificities of Groups 1,2,3, and 4 in the result section and not just in the figure legend of fig. 1. Co-relation between heating and the findings of Fig. 1 need to be explained better.

Response 3: Thank you for pointing this out. We agree that adding the specificities of the groups in the results section will enhance the clarity of their characteristics. We have added the specifications of the groups in the results first section (Line 91-95): “The colonies with those characteristics were classified into four categories according to the type of food from which they were isolated (Table S1): G1: sauces made from vegetables with or without heating procedures, G2: raw vegetables, G3: whole fruits, shakes, and juices, and G4: cooked meat in different presentations and tortillas, bread, and other food made with flour that underwent heating procedures”. We appreciate your suggestions for strengthening this section.

Comments 4: In Fig. 2: It would be better to address the inhibition halos as zone of clearance or zone of inhibition. Given that the authors are using the unit of "cm" to measure inhibition, it should be addressed as diameter and not area; especially when cm is used as the unit.

Response 4: Thank you for pointing this out. To address this, we have changed “inhibition halos” to “inhibition zone” in Fig. 2 legend and the graph's Y axis.

Comments 5: Throughout the draft, "sensibility" can be replaced with "sensitivity" to improve the scientific readability of this manuscript.

Response 5: Thank you for your observation. In response, we have replaced “sensibility” with “sensitivity” in lines 24, 162, 166, 188, 223, 322.

Comments 6: Overall, the manuscript is very dense on bar graphs and other graphically appealing ways can be used to present this data.

Response 6: Thank you for your observation. We have discussed and reflected on the option of presenting the results differently. However, we consider it essential to visualize the results by graphs since it allows a more facilitated, summarized, and simple understanding of E. coli contamination over the years, and the behavior of resistance against antibiotics. This even allows us to lead a discussion on the effectiveness of treatment against bacterial infection and highlight the most efficient antibiotics. If, on the other hand, we decide instead of graphs to use tables to validate the behavior of our variables, it would increase the data load in the document.

Comments 7: Please also outline the limitations of this study. 

Response 7: Thank you for pointing this out. We agree that addressing the study´s limitations is crucial. Accordingly, we have added a paragraph on lines 367-375 that addresses the limitations of this study. “While this work provides important information regarding the resistance behavior of bacteria present in food prepared and marketed under street conditions, there are some limitations that must be considered. Due to the dynamic factor of this type of food, it was difficult to collect samples from the same seller during the three years that our study lasted. There is a socio-economic impact that affects the presence or absence of vendors. However, we offer an approximation of the resistance phenomenon in this geographical area, collecting samples during the same season and at the same points of sale. Therefore, we provide a first insight into the state of resistance and its correlation with microbial load in food, laying the groundwork for further research”. We hope these changes meet your expectations. 

Reviewer 2 Report

Comments and Suggestions for Authors

The manuscript presents valuable findings on antibiotic-resistant Escherichia coli in street-vended food in Mexico but requires further refinement.

The methodology should clearly define the criteria for sample selection, including sample size calculation and selection bias mitigation. Statistical analysis would benefit from additional details on effect size and confidence intervals to improve result interpretation.

The discussion should include a direct comparison of resistance profiles with similar studies from other Latin American regions.

Finally, the conclusion should propose strategies for monitoring and mitigating antibiotic resistance, with recommendations for stricter hygiene regulations and alternative treatment options. Addressing these aspects would enhance the study’s clarity, impact, and practical relevance.

Author Response

Thank you very much for taking the time to review this manuscript. Please find the detailed responses below and the corresponding revisions in the re-submitted files.

Comments 1: The methodology should clearly define the criteria for sample selection, including sample size calculation and selection bias mitigation. Statistical analysis would benefit from additional details on effect size and confidence intervals to improve result interpretation.

Response 1: Thank you for pointing this out.  To address this, we have changed the methodology of sampling by adding (Line 380-386): “The three selected zones were located at “Mercado 16 de septiembre” (19.295329720085046, -99.65233073060944), “Mercado Juárez” (19.278670344550665, -99.6390527898009), and “Terminal de Autobuses” (19.277737071702695, -99.64292748169059); crowded areas with several food stalls by the roadside.  The food items were selected aleatory from stalls that matched the following characteristics: ready-to-eat food, exposure to vehicular traffic, and lack of proper infrastructure like access to safe water and refrigeration.” And lines 395-397: “The Raosoft sample size calculation method with a 5% margin of error and 95% confidence level was used to determine the sample size”. We trust this addition strengthens the manuscript.

Comments 2: The discussion should include a direct comparison of resistance profiles with similar studies from other Latin American regions.

Response 2: Thank you for your observation. We appreciate your suggestion to include a comparison between similar studies from other Latin American regions. In response, we added a paragraph that compared our results with two studies from Latin America: one from Portugal and the other from Colombia. Lines 317-324: “Studies from other Latin American regions exhibited differential antibiotic-resistant profiles. For example, in Portugal, E. coli strains isolated from food items displayed resistant profiles against ciprofloxacin, nalidixic acid, chloramphenicol, tetracycline, and TMP-SMX [43]; antibiotics to which the E. coli strains from this study were susceptible.  Likewise, Colombian Paipa cheese exhibited the presence of E. coli in 25% of the samples with sensitivity profiles against ampicillin, while revealed resistant profiles to TMP-SMX [44]. These results demonstrate the importance of national and local strategies to counter this problem “. Reference numbers were adjusted since two other references were added.

Comments 3: Finally, the conclusion should propose strategies for monitoring and mitigating antibiotic resistance, with recommendations for stricter hygiene regulations and alternative treatment options. Addressing these aspects would enhance the study’s clarity, impact, and practical relevance.

Response 3: Thank you for pointing this out. Accordingly, we added the following sentence (line 472-475): “These findings provide important information to local health centers, allowing medical specialists to analyze and determine better therapeutic strategies, highlighting the importance of strict hygienic regulations on food prepared and marketed on streets and dugs prescription control.”

Reviewer 3 Report

Comments and Suggestions for Authors
  1. The title of the manuscript is too long that it will be difficult for readers from first sight to understand the main topic of the manuscript. I suggest to shorten it and make it clear.
  2. Do authors have certificates to collect food samples and test them? If yes, I suggest to upload them.
  3. It seems that Figure 4S is created by using AI tools. I suggest to upload license of this tool.
  4. It will be better to describe samples geographical zones.
  5. Describing results are complicated, I suggest to write simple and clear.
  6. Manuscript is required to edit as in some sentences it is very difficult to understand the opinion of the authors.
  7. Maybe it will be better to differentiate the samples in the bar chart with the color. Because, it is little bit irregular.
Comments on the Quality of English Language

It is required to edit manuscript.

Author Response

Thank you very much for taking the time to review this manuscript. Please find the detailed responses below and the corresponding revisions in the re-submitted files.

Comments 1: The title of the manuscript is too long that it will be difficult for readers from first sight to understand the main topic of the manuscript. I suggest to shorten it and make it clear.

Response 1: We appreciate your comment; in fact, we have tried to design a title that, apart from reflecting the general objective, contains the keywords of our study, but due to its complex structure, we found some complications. When reviewing the state of the art and comparing it with other works, we consider that the title has essential aspects of the study. For example, it mentions the pathotypes found in food samples (extraintestinal and intestinal E. coli pathotypes); for many research groups, both are independent study subjects. We mention the region of the study, which facilitates the understanding of the geographical conditions, but if you consider it, we can omit it since it is mentioned in the methodology. Finally, the differential antibiotic resistance phenotype found among E. coli, which is the most important data of our study intentions.

Comments 2: Do authors have certificates to collect food samples and test them? If yes, I suggest to upload them.

Response 2: We appreciate your comment; it is very pertinent. Considering that our study does not analyze clinical samples or any direct metric on patients in our institution, we were not requested a document for review by the bioethics committee. No certificate was required when exploring food prepared and marketed in street conditions without sanitary regulations. On the other hand, the study was conducted with maximum confidentiality to obtain truthful data on the foods of the informal food trade. For further studies, where we intend to use the same scheme with bacteria isolated from clinical samples, we will follow a protocol established by the institutional bioethics committee.

Comments 3: It seems that Figure 4S is created by using AI tools. I suggest to upload license of this tool.

Response 3: Thank you for pointing this out. We agree it is crucial, and we have added the citation of the tool used to create this figure, BioRender. 

Comments 4: It will be better to describe samples geographical zones.

Response 4: Thank you for pointing this out. To address this, We have modified the description of the geographical zones by adding the following sentences in lines 380-386: “The three selected zones were located at “Mercado 16 de septiembre” (19.295329720085046, -99.65233073060944), “Mercado Juárez” (19.278670344550665, -99.6390527898009), and “Terminal de Autobuses” (19.277737071702695, -99.64292748169059); crowded areas with several food stalls by the roadside.  The food items were selected aleatory from stalls that matched the following characteristics: ready-to-eat food, exposure to vehicular traffic, and lack of proper infrastructure like access to safe water and refrigeration”. We hope these changes meet your expectations.

Comments 5: Describing results are complicated, I suggest to write simple and clear.

Response 5: Thank you for pointing this out. We understand that the description could be clearer. We have revised and re-wrote several sentences, making them more straightforward and precise. We have a document that tracks all the changes made; we will gladly provide it if you consider it necessary. We hope this improvement resolves any confusion.

Comments 6: Manuscript is required to edit as in some sentences it is very difficult to understand the opinion of the authors.

Response 6: Thank you for your comment. We appreciate your attention to detail and agree that the manuscript required some improvements to make it easier to understand. We have revised the manuscript and made several changes to clarify our opinions. We have a document that tracks all the changes made, and we will be happy to provide that document if you consider it necessary. We hope these changes address your concerns effectively.

Comments 7: Maybe it will be better to differentiate the samples in the bar chart with the color. Because, it is little bit irregular.

Response 7: Thank you for your observation. We have changed the format of the bars by adding colors to make it easier to understand and differentiate the data in Figures 1, 2, 3, and 4.  We appreciate your suggestions for strengthening this section.